# Depression and Anxiety Mediate the Association between Sleep Quality and Self-Rated Health in Healthcare Students

**DOI:** 10.3390/bs13020082

**Published:** 2023-01-19

**Authors:** Yihong Zhu, Chen Jiang, You Yang, Joseph M. Dzierzewski, Karen Spruyt, Bingren Zhang, Mengyi Huang, Hanjie Ge, Yangyang Rong, Bolanle Adeyemi Ola, Tingjie Liu, Haiyan Ma, Runtang Meng

**Affiliations:** 1School of Clinical Medicine, Hangzhou Normal University, Hangzhou 311121, China; 2School of Public Health, Hangzhou Normal University, Hangzhou 311121, China; 3School of Nursing, Hangzhou Normal University, Hangzhou 311121, China; 4The National Sleep Foundation, Washington, DC 20036, USA; 5Université de Paris, NeuroDiderot, INSERM, 75019 Paris, France; 6Department of Behavioral Medicine, Faculty of Clinical Sciences, Lagos State University College of Medicine, Ikeja, Lagos 21266, Nigeria; 7Engineering Research Center of Mobile Health Management System, Ministry of Education, Hangzhou 311121, China

**Keywords:** sleep quality, self-rated health, anxiety, depression, mediation, healthcare students

## Abstract

Objectives: This study aimed to investigate factors associated with sleep quality in healthcare students and to determine whether depressive and anxiety symptoms may explain some of the associations between sleep quality and self-rated health. Study design: This is a cross-sectional study at wave one. Methods: A total of 637 healthcare students were recruited via a stratified random sampling method in Hangzhou, China. The Sleep Quality Questionnaire (SQQ) and the four-item Patient Health Questionnaire (PHQ-4) were used to assess sleep quality, depressive, and anxiety symptoms, respectively. Self-rated health was assessed via a self-developed questionnaire of both physical and psychological health. Structural equation modeling was used to examine the direct and indirect effects of sleep quality on self-rated health through depressive and anxiety symptoms. Results: Students engaged in part-time employment (*p* = 0.022), who had poor perceived employment prospects (*p* = 0.009), and who did not participate in recreational sports (*p* = 0.008) had worse sleep quality. Structural equation modeling revealed a significant total effect of sleep quality on self-rated health (*b* = 0.592, *p* < 0.001), a significant direct effect of both sleep quality and depressive and anxiety symptoms on self-rated health (*b* = 0.277, 95% CI: 0.032–0.522), and a significant indirect effect of sleep quality on self-rated health through depressive and anxiety symptoms (*b* = 0.315, 95% CI: 0.174–0.457). Conclusions: Depressive and anxiety symptoms partially explain the association between sleep quality and self-rated health. Intervening upon sleep quality, depressive, and anxiety symptoms may bolster the self-rated health of healthcare students.

## 1. Introduction

Sleep disorders have a global burden. Prior research has reported the prevalence of any sleep disorders to be 36.730% in the general population, 41.160% in college students, and even higher in healthcare students [1,2]. A meta-analysis of 50 studies that used the Pittsburgh Sleep Quality Index (PSQI) to quantify sleep quality revealed that the overall prevalence of poor sleep quality among healthcare student cohorts was 52.700% [2].

Previous studies have demonstrated a significant association between stress and low sleep quality—for healthcare students, and the main source of stress is academic demands [3]. Compared with college students in other specialties/majors, healthcare students are exposed to unique stressors and demands, including tough courses, demanding course content, long tenure in school, lack of leisure time, and excessive self-pressure for performance, all of which can lead to high levels of stress and low levels of sleep quality [4,5]. Moreover, due to the outbreak of the Coronavirus disease 2019 (COVID-19), several outbreak prevention and control measures (e.g., work stoppages, school closures, and closure management) ensued, all of which may have a negative impact on sleep among university students [6,7]. Not only that, due to lockdowns, going out for recreation or exercising became difficult [8]. Therefore, people are more likely to relax by playing games, watching videos, and using social media software, with sedentary behavior increasing substantially [9]. The extended use of electronic products also increases the risk of addiction. It has been shown that individuals addicted to the Internet have significantly less sleep time and lower sleep quality [10]. At the same time, the levels of stress, anxiety, and depression may also have enhanced which may have a significant impact on health [11,12].

Sleep, a biological necessity, has clear relevance for self-rated health [13]. The World Health Organization defines “health” as an optimal state of physical, mental, emotional, intellectual, and social being [14]. Common disease states which interfere with achieving optimal health, such as diabetes and obesity, are very prevalent worldwide in both adults and children [15]. Excessive or short sleep duration, difficulty falling asleep, and low levels of slow-wave sleep all are associated with an increased risk of type 2 diabetes [16,17]. A meta-analysis showed an increased risk of atherosclerosis, coronary heart disease, hypertension, and stroke in individuals with both short and excessively long sleep duration [18,19]. These are only some of the many examples of disrupted sleep being associated with reduced health status.

Disturbed sleep has been implicated in numerous psychiatric conditions (e.g., schizophrenia, affective disorders, depression, and Alzheimer’s disease) [20,21,22]. In fact, sleep disturbances have been reported in 30–80% of patients with schizophrenia, and improving sleep is one of the top reported priorities among people with schizophrenia [23]. Up to 90% of patients experiencing an acute depressive episode also report persistent sleep difficulties. Interestingly, clinical research suggests that changes in sleep may precede the occurrence of anxiety and depression, and not the reverse [24,25]. General college students have high rates of anxiety and depression, with even higher rates reported among healthcare students [26].

There is a strong rationale for greater attention to the sleep quality of healthcare students, as they will become essential healthcare workers in the near future. Due to socio-economic and cultural differences, the extant literature on sleep quality, depressive and anxiety symptoms, and self-rated health may not be applicable to individuals in China. The lack of applicable data makes it difficult to propose possible interventions to improve sleep quality among Chinese healthcare students [27]. Additionally, there is a dearth of research investigating how depressive and anxiety symptoms may explain the association between sleep quality and self-rated health among healthcare students. Hence, we aimed to investigate the following problems among Chinese healthcare students: (a) if sleep quality was associated with self-rated health, and (b) if depressive and anxiety symptoms explained any observed associations between sleep quality and self-rated health. We hypothesized that: (i) sleep quality would be positively associated with self-rated health, and (ii) depressive and anxiety symptoms would mediate the association between sleep quality and self-rated health. In other words, we investigated whether good sleep quality may reduce depressive and anxiety symptoms in healthcare students, which in turn may increase self-rated health.

## 2. Materials and Methods

### 2.1. Study Design and Procedure

The present study employed an observational, cross-sectional design. Surveys were applied using a paper-and-pencil self-administered method, and all design elements adhered to the STrengthening the Reporting of OBservational studies in Epidemiology (STROBE) guidelines [28]. Healthcare students were recruited using a stratified random sampling method according to matriculation year and major from Hangzhou, China, during the fall semester of 2020. Data were collected during a low-risk COVID-19 infection period in the region of the school.

Data collection was conducted during breaks between classes, and all students who reported to class on data collection days were included. The exclusion criteria were as follows: (1) international students who could not fully understand and write Chinese; (2) students who were on suspension or long-term medical leave. Well-trained investigators notified potential participants of the research purpose before the survey, and verbal informed consent was obtained from all individual participants included in the study. All surveys were scanned for missing data upon completion, and any identified missing data were completed before the participant was compensated for time. Each participant received CNY 2 (around USD 0.300) upon completion of the survey as compensation for their time. Participants could withdraw from the study at any point.

### 2.2. Measures

#### 2.2.1. Sleep Quality Questionnaire (Chinese Version)

The Sleep Quality Questionnaire (SQQ) was developed to assess sleep quality in non-clinical populations over the previous month. It is composed of 10 items and two subscales: Daytime Sleepiness Subscale (DSS) (e.g., “I sometimes felt sleepy during the day”) and Sleep Difficulty Subscale (SDS) (e.g., “I felt like I did not get a deep sleep”) [29]. Responses are reported on a five-point Likert scale (from “0 = strongly agree” to “4 = strongly disagree”), with total scores ranging from 0 to 40, with higher scores indicating poorer sleep quality. The SQQ has shown good measurement properties (i.e., reliability, internal consistency, construct validity, criterion validity, and content validity) and stable two-factor structure (i.e., measurement invariance) in large multicenter studies [29,30,31,32,33].

#### 2.2.2. Patient Health Questionnaire (Chinese Version)

The four-item Patient Health Questionnaire (PHQ-4), composed of the two-item Patient Health Questionnaire and the two-item Generalized Anxiety Disorder scale (GAD), is used to assess depressive and anxiety symptoms during the past two weeks [34]. Participants rated how often they experienced each of the four items on a four-point Likert scale ranging from 0 (“not at all”) to 3 (“nearly every day”), with higher scores indicating more depressive and anxiety symptoms [35]. The Chinese version of the PHQ-4 (retrieved from: https://www.phqscreeners.com; accessed on 29 August 2019) has shown acceptable internal consistency [Cronbach’s α = 0.870 (95% CI: 0.852–0.886)] [30].

#### 2.2.3. Self-Rated Health Questionnaire

Self-rated health was assessed with a simple, two-item questionnaire that included an item focused on self-rated physical health and an item focused on self-rated mental health. Each item response was recorded on a five-point Likert scale (“1 = excellent, 2 = good, 3 = average, 4 = poor, and 5 = extremely poor”). Total scores represent the numerical sum of both items and range from 2 to 10, with higher scores representing poorer overall self-rated health. The Cronbach’s α of the Self-Rated Health Questionnaire (SRHQ) was 0.706 in this wave.

#### 2.2.4. Sample Description

The following sample variables were collected: gender (male, female), age (<20, ≥20), home location (urban, rural, or suburban), only child (yes, no), academic year (first year, second year, or third year), family income (<CNY 10,000, ≥CNY 10,000; unit: CNY 1 ≈ USD 0.150), part-time employment (yes, no), leisure time sports involvement (yes, no), perceived employment prospects (excellent, good, average, poor, or extremely poor), engagement in hobbies (yes, no), preferred coping strategies (actively cope, push through, or ignore problems), and academic major (clinical medicine, preventive medicine, nursing, pharmacy, health policy and management, health services and management, or master of medicine).

### 2.3. Statistical Analysis

The database was built in EpiData (version 3.1) software. Statistical analysis was conducted with R (version 4.1.2) and JASP (version 0.16.1). For missing values (0.229%), continuous variables were replaced by mean (i.e., age) and categorical variables were replaced by median (e.g., gender, home location, and academic year) [36]. A one-sample *t*-test and one-way analysis of variance (ANOVA) were used to compare differences in the SQQ scores between subgroups (e.g., academic year and part-time employment). Spearman’s correlation was used to evaluate the association among the SQQ, PHQ-4, and SRHQ.

A structural equation model (SEM) was used to model the direct, indirect, and total effects of sleep quality on self-rated health, and whether depressive and anxiety symptoms explained any observed associations. First, the direct effects of sleep quality on self-rated health were estimated. If the direct effects were significant, the mediating variables (i.e., depressive and anxiety symptoms) were added to the model, allowing for analyses of indirect effects and total effects. A bias-corrected (percentile method) bootstrap procedure (1000 bootstrap procedures) was used to estimate the model parameters. To assess the goodness-of-fit of the models, we used the following threshold values of fit indices: Chi-square/degree of freedom (*χ^2^*/*df*): 2.000–3.000, Root Mean Square Error of Approximation (RMSEA) < 0.080, Standardized Root Mean Residual (SRMR) < 0.080, Goodness of Fit Index (GFI) > 0.900, Tucker–Lewis Index (TLI) > 0.900, Comparative Fit Index (CFI) > 0.900, Adjusted Goodness-of-Fit Index (AGFI) > 0.500, and Parsimony Normed Fit Index (PNFI) > 0.500 [37,38,39]. 

## 3. Results

### 3.1. Sample Characteristics

After removing invalid questionnaires (e.g., selecting one option for the entire page of the questionnaire), the final sample was 637. Three-quarters of the respondents were female, and 40% of the students were an only child. The number of those who had no part-time employment (82.732%) was significantly higher than the number who did have outside employment (17.268%). A total of 84.144% of the respondents had a “good” and “average” attitude toward employment prospects. Worse sleep quality was found in students who were: less than 20 years of age (*p* = 0.007); sophomores (*p* = 0.034); participated in part-time jobs (*p* = 0.022); did not engage in leisurely sports (*p* = 0.008); had relatively low perceived employment prospects (*p* = 0.009); had no hobbies (*p* = 0.014); and preferred ignoring problems as a coping strategy (*p* < 0.001) (Table 1).

### 3.2. Associations among Sleep Quality, Mental Health Symptoms, and Self-Rated Health

The Spearman’s correlation coefficients for scores among the SQQ, PHQ-4, and SRHQ were all clustered around 0.400 and 0.500. The coefficients between the scores of the SQQ and PHQ-4 (*r* = 0.476), between the PHQ-4 and SRHQ (*r* = 0.465), and between the SQQ and SRHQ (*r* = 0.406) were all of the moderate/large magnitudes. Please see Figure 1 for a full list of the correlation coefficients.

### 3.3. Structural Equation Model: Direct, Indirect, and Total Effects of Sleep Quality on Self-Rated Health through Mental Health Symptom

The total effect of sleep quality on self-rated health was 0.592 (*p* < 0.001) (Figure 2A). Standardized regression coefficients between sleep quality, depressive and anxiety symptoms, and self-rated health are displayed in Figure 2B. Statistical significant paths were observed between sleep quality and depressive and anxiety symptoms (*a* = 0.704), and between depressive and anxiety symptoms and self-rated health (*b* = 0.448). The direct effect between sleep quality and self-rated health, accounting for depressive and anxiety symptoms, was 0.227 (95% CI: 0.032–0.522, *p* < 0.050). The indirect effect of sleep quality on self-rated health through depressive and anxiety symptoms was 0.315 (95% CI: 0.174–0.457, *p* < 0.010). The fit of the model was excellent (RMSEA = 0.069, SRMR = 0.027, GFI = 0.987, *χ^2^/df* = 4.075, TLI = 0.959, CFI = 0.984, AGFI = 0.955, and PNFI = 0.391), except for the PNFI of 0.391 which was slightly below the threshold (Table 2).

## 4. Discussion

Our study had three main findings: (i) depressive and anxiety symptoms, and a host of descriptive characteristics (such as perceived employment prospects, academic year, family income, and leisure time sports involvement) were associated with sleep quality; (ii) associations among sleep quality, depressive and anxiety symptoms, and self-rated health all were statistically significant (and of moderate/strong strength); and (iii) sleep quality was positively associated with self-rated health; depressive and anxiety symptoms mediated the relationship between sleep quality and self-rated health.

Lower sleep quality among healthcare students has been well-documented and the present study replicated these previous findings [40,41]. The results showed that sleep quality was lower for an individual with poor perceived employment prospects. Unstable working conditions and unemployment may disrupt sleep by promoting anxiety about the future (i.e., insecurity) [42,43]. Raising the employment expectations of those located at the margins of the labor market may be beneficial in preventing the widening of employment-related health inequalities [44]. Although participation in part-time employment and sleep outcomes have been understudied, a German study found that men with part-time employment were more likely to have sleep difficulties than individuals with full-time employment [45]. Whether part-time employment itself or other factors are responsible for the decrease in sleep quality is unknown, and further research is needed. We have also found that engagement in leisure time physical activity was associated with sleep quality. Improvement in sleep quality resulting from exercise is well documented. Moderate aerobic exercise training is recommended as a non-pharmacological treatment option for people with poor sleep [46].

Consistent with previous research, poorer sleep quality was associated with depressive and anxiety symptoms. Previous research has shown that sleep disturbance is a strong predictor of subsequent affective disorders, anxiety, and suicide in longitudinal studies [47]. Importantly, the reduction in sleep problems has been shown to be effective in reducing the incidence of depression at one-year follow-ups [48]. As such, a body of evidence now suggests that sleep disorders often occur prior to depressive and anxiety symptoms and may be a prodromal symptom of subsequent mental disorders. Timely intervention for sleep disorders may prevent the subsequent onset or exacerbation of depressive and anxiety symptoms. Consistent with our findings, prevention and intervention of sleep disorders may be the most appropriate and cost-effective way to reduce the rate of depression and anxiety [49].

The effect of sleep quality on self-rated health was shown to be partially mediated through depressive and anxiety symptoms. Importantly, self-rated health is a good measure of objective and subjective health and has documented high reliability [50]. Self-rated health has also been shown to be significantly associated with many important medical endpoints (e.g., health risk behaviors, disease states, disability, and mortality) [50]. Healthcare students, a group with increased levels of medical knowledge and relatively high sensitivity to physical conditions, may be more likely to perceive their own physical and mental health conditions—they may be more self-aware [51]. Previous research has shown that people with psychological complaints tend to have lower ratings of their health status [52,53]. Numerous previous studies have shown that common psychological problems (e.g., depression and anxiety) are associated with self-rated health [54,55,56,57,58]. Novel to our current study, we found that depressive and anxiety symptoms partially explain the link between sleep and self-rated health. Although causal inferences cannot be drawn from this study directly, the role of depressive and anxiety symptoms in the link between sleep quality and self-rated health is noteworthy. Future work should examine methods to improve sleep in healthcare students as a means to also target associated mental health and physical health conditions [59].

### 4.1. Strengths and Limitations

The present study has several strengths. The survey was conducted in person, allowing for visual confirmation that study participants were drawn from the desired subject pool. Additionally, we were able to quickly scan paper documents to identify and address sources of potential missing data. The sample size of the study was large and included a diverse set of healthcare students. Lastly, this study examines a potential process, namely, depressive and anxiety symptoms, through which poor sleep quality may be exerting its effects on self-rated health. This is an important step in moving the literature forward.

There are also several disadvantages of the present investigation. First, given the study design, it is not possible to draw causal conclusions. Second, all data were obtained through subjective assessments, as opposed to objective measures, which has the potential to result in a self-reporting bias. Third, the SRHQ, as a self-developed questionnaire, needs further validation for its measurement properties. Fourth, the survey was conducted during COVID-19, but no COVID-19-related information was collected to understand the current impact of COVID-19 policy on healthcare students. Finally, as this study only enrolled healthcare students, the sample is relatively homogeneous and the study findings may not generalize to non-healthcare student samples.

### 4.2. Future Directions

Subsequent studies would be well suited to collect longitudinal measurements in order to further elucidate causality. Longitudinal data are necessary to model the complex interplay of dynamic changes in sleep quality, depressive and anxiety symptoms, and health. Secondly, our results suggest partial mediation of the effects of sleep quality on self-rated health through depressive and anxiety symptoms, suggesting the presence of other mediators that should be further explored [60]. Lastly, future research should continue to recruit increasingly diverse samples to ensure that our studies apply to traditionally under-represented individuals.

## 5. Conclusions

Sleep quality is associated with self-rated health and this association partially works through increased levels of depressive and anxiety symptoms. Healthcare students face numerous demands and experience poor sleep, heightened levels of depressive and anxiety symptoms, and low self-rated health. Sleep health education programs and routine screening with the SQQ may be possible mitigation strategies. Future research should examine longitudinal changes in sleep quality, depressive and anxiety symptoms, and self-rated health, as well as begin the process of development and initial testing of potential sleep health interventions. Compromises in the health of future healthcare providers could have dire consequences for large swaths of the population.

## Figures and Tables

**Figure 1 behavsci-13-00082-f001:**
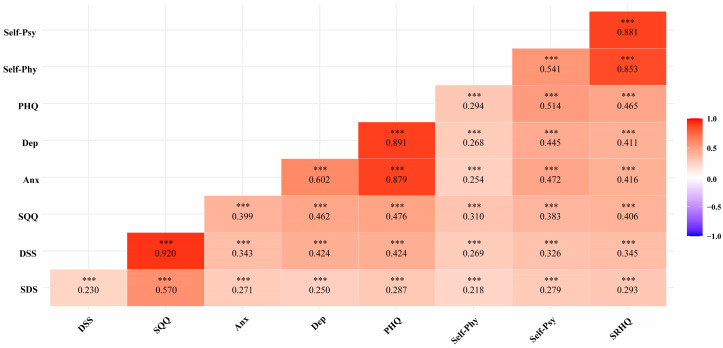
Spearman’s correlation matrix of the SQQ, PHQ, and SRHQ. Note: SDS, Sleep Difficulty Subscale; DSS, Daytime Sleepiness Subscale; SQQ, Sleep Quality Questionnaire; Anx, Anxiety; Dep, Depression; PHQ, Patient Health Questionnaire; Self-Phy, Self-Rated Physical Condition; Self-Psy, Self-Rated Psychological Condition; SRHQ, Self-Rated Health Questionnaire; *** *p* < 0.001.

**Figure 2 behavsci-13-00082-f002:**
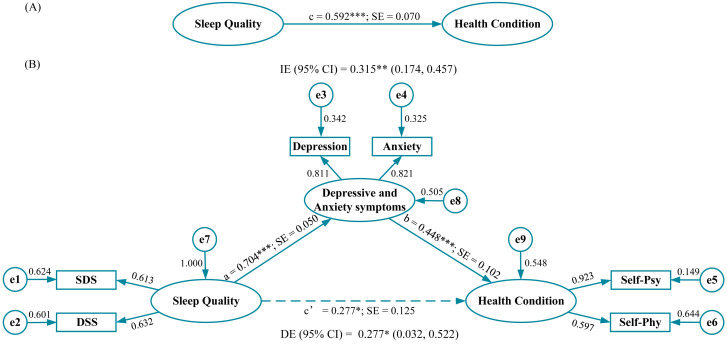
Path coefficients for simple mediation analysis on sleep quality (*n* = 637). (**A**) The total effect of sleep quality on health condition and (**B**) the effect of sleep quality on self-rated health when anxiety and depression are included as a mediator. Note: Dotted line represents the effect of sleep quality on health condition when anxiety and depression are included as a mediator. Factor loadings are standardized. a, b, c, and c’, standardized regression coefficients; e, measurement errors; SDS, Sleep Difficulty Subscale; DSS, Daytime Sleepiness Subscale; Self-Psy, Self-Rated Psychological Condition; Self-Phy, Self-Rated Physical Condition; SE, Standard Error; IE, Indirect Effect; DE, Direct Effect; *** *p* < 0.001; ** *p* < 0.010; * *p* < 0.050.

**Table 1 behavsci-13-00082-t001:** Characteristics of participants (*n* = 637).

Variable	*n* (%)	SQQ Total		Statistics		
		Mean	SD	*df*	*t/F*	*p*
Gender				1	2.382	0.123
Male	156 (24.490)	17.462	6.539			
Female	481 (75.510)	18.345	6.104			
Age				1	7.326	0.007
<20	458 (71.900)	18.544	6.187			
≥20	179 (28.100)	17.067	6.194			
Home location				2	2.528	0.081
Urban	246 (38.619)	17.622	6.064			
Rural	236 (37.049)	18.055	6.300			
Suburban	155 (24.333)	19.045	6.280			
Only child				1	2.396	0.122
Yes	257 (40.345)	17.665	6.273			
No	380 (59.655)	18.442	6.172			
Academic year				2	3.386	0.034
First year	274 (43.014)	17.810	6.464			
Second year	161 (25.275)	19.224	5.967			
Third year	202 (31.711)	17.688	6.002			
Family income				1	0.040	0.842
<CNY 10,000	261 (40.973)	18.188	6.205			
≥CNY 10,000	376 (59.027)	18.088	6.238			
Part-time employment				1	5.276	0.022
Yes	110 (17.268)	19.364	5.967			
No	527 (82.732)	17.871	6.246			
Leisure time sports involvement				1	7.068	0.008
Yes	316 (49.608)	17.472	6.314			
No	321 (50.392)	18.776	6.067			
Perceived employment prospects				4	3.431	0.009
Excellent	56 (8.791)	17.982	7.129			
Good	347 (54.474)	17.715	5.996			
Average	189 (29.670)	18.228	6.340			
Poor	39 (6.122)	21.564	5.389			
Extremely poor	6 (0.942)	18.000	6.099			
Engagement in hobbies				1	6.133	0.014
Yes	451 (70.801)	17.738	6.338			
No	186 (29.199)	19.075	5.831			
Preferred coping strategies				2	22.558	<0.001
Active copy	375 (58.870)	18.664	5.793			
Push through	211 (33.124)	16.237	6.240			
Ignore problems	51 (8.006)	22.020	6.701			
Academic major				6	1.483	0.181
Clinical medicine	127 (18.937)	18.378	6.279			
Preventive medicine	98 (15.385)	18.612	5.452			
Nursing	93 (14.600)	18.774	6.216			
Pharmacy	95 (14.914)	18.789	5.909			
Health policy and management	87 (13.658)	17.471	6.077			
Health services and management	76 (11.931)	17.566	6.616			
Master of Medicine	61 (9.576)	16.459	7.208			

SD, Standard Deviation; *df*, degrees of freedom; SQQ Total, Total score of the Sleep Quality Questionnaire; Family income unit: CNY 1 ≈ USD 0.150.

**Table 2 behavsci-13-00082-t002:** Evaluation of the goodness-of-fit of the models (*n* = 637).

GOF Index	Mediation Model	Threshold
Absolute measures		
RMSEA (90% CI)	0.069 (0.042, 0.099)	<0.080
SRMR	0.027	<0.080
GFI	0.987	>0.900
*χ*^2^/*df*	4.075	2.000–3.000
Incremental fit measures		
TLI	0.959	>0.900
CFI	0.984	>0.900
Parsimony measures		
AGFI	0.955	>0.900
PNFI	0.391	>0.500

GOF, goodness-of-fit; RMSEA, Root Mean Square Error of Approximation; SRMR, Standardized Root Mean Residual; GFI, Goodness-of-Fit Index; *χ*^2^, Chi-square; *df*, degree of freedom; TLI, Tucker–Lewis Index; CFI, Comparative Fit Index; AGFI, Adjusted Goodness-of-Fit Index; PNFI, Parsimony Normed Fit Index; CI, Confidence Interval.

## Data Availability

Not applicable.

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
