# Peer review of "Depression and Anxiety Mediate the Association between Sleep Quality and Self-Rated Health in Healthcare Students"

_behavsci, 2023, doi:10.3390/bs13020082_

Round 1
Reviewer 1 Report
1. Statistical analysis: t test may not be suitable for the analysis as the data was probably not normal-distributed. The statistical analysis should be carefully reviewed.
2. There were no clear inclusion and exclusion criterias.
3. The medical history of the participants should be investigated.
4. I wonder if there was more objective questionnaire to assess the health of the participants than self-rated health questionnaire.
Author Response
Responses to Reviewer Comments:
Reviewer #1:
We would like to thank the reviewer for your time and efforts in reviewing our article.
- Statistical analysis: t test may not be suitable for the analysis as the data was probably not normal-distributed. The statistical analysis should be carefully reviewed.
We re-run the normality test and found that the data are normally distributed. We have removed the misrepresentation of non-normal distribution in the paper. Additionally, we have also double-checked other parts of the data analysis.
Line 178 – 179, page 4: A bias-corrected (percentile method) bootstrap procedure (1,000 bootstrap procedures) was used to estimate model parameters.
- There were no clear inclusion and exclusion criterias.
Thank you for your constructive comment. Inclusion and exclusion criteria were added.
Line 115 – 118, page 3: Data collection was conducted during breaks between classes, and all students who reported to class on data collection days were included. The exclusion criteria were as follows: 1) international students who could not fully understand and write Chinese; 2) students who were on suspension or long-term medical leave.
- The medical history of the participants should be investigated.
We thank the reviewer for the suggestion. This study is an investigation of the general status of sleep quality among healthcare students in Hangzhou, China, and we do not focus on the special status of those with a medical history. One might furthermore presume that students attending on a regular basis their classes, are without serious health issues that may potentially bias our sample of the subject under investigation.
In addition, to minimize the data collection burden for each respondent , ultimately, we avoided detailed investigation of their the medical history.
- I wonder if there was more objective questionnaire to assess the health of the participants than self-rated health questionnaire.
We agree with this insightful comment. Few scales could be applied for straightforwardly self-rating health status, as a consequence we chose to use a two-item questionnaire. In line 152, we added the reliability and validity indices of the questionnaire. We acknowledge that the reliability and validity obtained by a one single measure may have less credibility. Therefore, we add this as an additional limitation in the discussion section.
Line 152, page 4: The Cronbach’s α of the questionnaire was 0.706.
Line 311 – 312, page 10: Third, the SRHQ, as a self-developed questionnaire, needs further validation for its measurement properties.

Reviewer 2 Report
Assessing sleep quality, self-rated health, anxiety, and depression, the present work highlight the mediation role of depression and anxiety in the association between sleep quality and self-rated health in healthcare student.
Despite the above-reported variables being widely studied in the student population, this paper has the novelty of analyzing how these variables interact in influencing students' health.
Nevertheless, there are some aspects that should be clarified before making the work ready for publication.
1. First of all, before asking questions directly related to the content of the manuscript. The authors referred to collect data during fall 2021 in Hangzhou, China. My question is, there was restrictive measure (or similar things) to counteract covid-19 pandemic in fall 2021 in Hangzhou, China?
In 2021 the world population was still facing the pandemic. In the present paper, there are no references to covid/pandemic the presence of covid 19 infection in the month before the assessment or similar things.
I am really interested in why authors have chosen not to refer to any of these conditions. Despite the literature is full of works that have highlighted how covid has compromised the already precarious condition of students (e.g. https://doi.org/10.3390/ijerph182413346 or https://doi.org/10.1371/journal.pone.0236337).
2. Lines 81 to 83: I kindly invite the authors to insert the appropriate citation.
3. Sentence " Hence, we aimed to investigate the following among Chinese healthcare students" (line 83-84) miss the subject. "Investigate the following"...what?. I know what the authors would like to say is clear reading the whole period. However, I invite the authors to enter the subject to make the sentence easier to understand.
4. Section 2.2.4: The authors collected a huge amount of variables (correctly), but I’m still here wondering why nothing is being asked about covid. Despite the well-known influence it has had on the lives of all of us.
5. Line 172: "and 60% of the students were the only child" I suppose the authors make a typing error. If I read table 1 it is reported that the students' only children are 40%, not 60%. Please correct the typo.
6. Lines 175 to 179: I kingly suggest the authors reformulate the sentence to sand a more clear message and more easily understandable. Maybe "Worse sleep quality was found in students who are less than 20 years of age (p = 0.007), who are sophomores (p = 0.034), who participated in part-time jobs (p = 0.022), who did not engage in leisurely sports (p = 0.008), who had relatively low perceived employment prospects (p = 0.009), who had no hobbies (p = 0.014), and who's preferred coping strategies is to ignore the problems (p < 0.001) (Table 1)." It is more direct and understandable with the object in the first part of the sentence.
7. Line 199: there is a typo, please insert the point (.) after figure 2B.
Author Response
Reviewer #2:
Assessing sleep quality, self-rated health, anxiety, and depression, the present work highlight the mediation role of depression and anxiety in the association between sleep quality and self-rated health in healthcare student.
Despite the above-reported variables being widely studied in the student population, this paper has the novelty of analyzing how these variables interact in influencing students' health.
Thank you very much for your detailed comments on our work. We believe that the feedback from the reviewers has strengthened our manuscript. We have responded to each of your comments individually below.
Nevertheless, there are some aspects that should be clarified before making the work ready for publication.
- First of all, before asking questions directly related to the content of the manuscript. The authors referred to collect data during fall 2021 in Hangzhou, China. My question is, there was restrictive measure (or similar things) to counteract covid-19 pandemic in fall 2021 in Hangzhou, China?
In 2021 the world population was still facing the pandemic. In the present paper, there are no references to covid/pandemic the presence of covid 19 infection in the month before the assessment or similar things.
I am really interested in why authors have chosen not to refer to any of these conditions. Despite the literature is full of works that have highlighted how covid has compromised the already precarious condition of students (e.g. https://doi.org/10.3390/ijerph182413346
or https://doi.org/10.1371/journal.pone.0236337).
Thank you for this insightful comment. In the fall of 2021 in Hangzhou, China, there were measures against COVID-19. For example, “5-day centralized quarantine and 3-day home quarantine” for COVID-19 close contacts; the areas where infected people live or have frequent activities are divided into “high- and low-risk areas” according to the situations (e.g., the high-risk areas are strictly quarantined). Consequently, the overall outbreak was at a manageable stage. There were no new indigenous cases in the area where the investigated college is located, so the health risk from COVID-19 to the participants was low during the period of data collection.
The two references mentioned by the reviewer describe changes in sleep quality, mental health status of Italian and Swiss students before and during the epidemic. We have studied carefully and added it to our manuscript. We are sorry for not describing the status of the Chinese epidemic in the previous manuscript. It is noteworthy that although the first COVID-19 infected population was found in our country, the prevalence has been relatively well controlled.
However, the direct or indirect impact of the outbreak on healthcare students cannot be denied. The present study focused on exploring the role of anxiety and depression between sleep quality and self-rated health rather than the effect of COVID-19 on sleep quality and mental health of healthcare students. Furthermore, given that the status of these participants prior to the outbreak was not collected, it is difficult to draw powerful conclusions about the negative impact of the outbreak on healthcare students based on a single survey. Certainly, it was an oversight that we did not collect information about COVID-19 at that time, and perhaps because COVID-19 was not our objective, so we also mentioned this in the limitations. Nonetheless, this historical event will likely influence all scientifically collected data, yet its external validity might be minimally affected given that it was a pandemic.
Incorporating your comment, we have made the following modifications:
Line 58 – 68, page 2: Moreover, due to the outbreak of the Corona Virus Disease 2019 (COVID-19), a number of outbreak prevention and control measures (e.g., work stoppages, school closures, and closure management) ensued, all of which may have a negative impact on sleep among university students.[6, 7] Not only that, due to the lockdown, going out for recreation or exercising has become difficult.[8] Therefore, people are more likely to relax by playing games, watching videos, and using social media software, with sedentary behavior increasing substantially.[9] The extended use of electronic products also increases the risk of addiction. It has been shown that individuals addicted to the Internet have significantly less sleep time and lower sleep quality.[10] At the same time, the level of stress, anxiety, and depression may also have enhancedwhich may have a signficant impact on health.[11, 12]
Line 312 – 314, page 10: Fourth, the survey was conducted during COVID-19, but no COVID-19-related information was collected to understand the current impact of COVID-19 policy on healthcare students.
References
- Yin, F.; Chen, C.; Song, S.; Chen, Z.; Jiao, Z.; Yan, Z.; Yin, G.; Feng, Z., Factors Affecting University Students’ Sleep Quality during the Normalisation of COVID-19 Epidemic Prevention and Control in China: A Cross-Sectional Study. Sustainability 2022; Vol. 14.
- Viselli, L.; Salfi, F.; D’Atri, A.; Amicucci, G.; Ferrara, M., Sleep Quality, Insomnia Symptoms, and Depressive Symptomatology among Italian University Students before and during the Covid-19 Lockdown. Int J Env Res Pub He 2021; Vol. 18.
- Kowalsky, R. J.; Farney, T. M.; Kline, C. E.; Hinojosa, J. N.; Creasy, S. A., The impact of the covid-19 pandemic on lifestyle behaviors in U.S. college students. J Am Coll Health 2021, 1-6.
- Meyer, J.; McDowell, C.; Lansing, J.; Brower, C.; Smith, L.; Tully, M.; Herring, M., Changes in Physical Activity and Sedentary Behavior in Response to COVID-19 and Their Associations with Mental Health in 3052 US Adults. Int J Env Res Pub He 2020, 17, (18), 6469.
- Alimoradi, Z.; Lin, C.-Y.; Broström, A.; Bülow, P. H.; Bajalan, Z.; Griffiths, M. D.; Ohayon, M. M.; Pakpour, A. H., Internet addiction and sleep problems: A systematic review and meta-analysis. Sleep Med Rev 2019, 47, 51-61.
- Zhong, Y.; Ma, H.; Liang, Y. F.; Liao, C. J.; Zhang, C. C.; Jiang, W. J., Prevalence of smartphone addiction among Asian medical students: A meta-analysis of multinational observational studies. Int J Soc Psychiatry 2022, 68, (6), 1171-1183.
- Elmer, T.; Mepham, K.; Stadtfeld, C., Students under lockdown: Comparisons of students' social networks and mental health before and during the COVID-19 crisis in Switzerland. PLoS One 2020, 15, (7), e0236337-e0236337.
- Lines 81 to 83: I kindly invite the authors to insert the appropriate citation.
We have added the reference for this statement.
References
- Sun, Y.; Wang, H.; Jin, T.; Qiu, F.; Wang, X., Prevalence of Sleep Problems Among Chinese Medical Students: A Systematic Review and Meta-Analysis. Front Psychiatry 2022, 13.
- Sentence " Hence, we aimed to investigate the following among Chinese healthcare students" (line 83-84) miss the subject. "Investigate the following"...what?. I know what the authors would like to say is clear reading the whole period. However, I invite the authors to enter the subject to make the sentence easier to understand.
We have added the subject as suggested by the reviewer.
Line 97 – 101, page 2 – 3: Hence, we aimed to investigate the following problems among Chinese healthcare students: (a) if sleep quality was associated with self-rated health, and (b) if depressive and anxiety symptoms explained any observed associations between sleep quality and self-rated health.
- Section 2.2.4: The authors collected a huge amount of variables (correctly), but I’m still here wondering why nothing is being asked about covid. Despite the well-known influence it has had on the lives of all of us.
Thank you for your comment. COVID-19 is not the core focus of the research yet we still put it into the limitations. Furthermore, as mentioned, COVID-19 will have a historical impact on scientific work, whether or not it is the topic of investigation.
Line 312 – 314, page 10: Fourth, the survey was conducted during COVID-19, but no COVID-19-related information was collected to understand the current impact of COVID-19 policy on healthcare students.
- Line 172: "and 60% of the students were the only child" I suppose the authors make a typing error. If I read table 1 it is reported that the students' only children are 40%, not 60%. Please correct the typo.
We have changed “60%” to “40%”, thanks for ccapturing this typo.
Line189 – 190, page 4: Three-quarters of the respondents were female, and 40% of the students were the only child.
- Lines 175 to 179: I kingly suggest the authors reformulate the sentence to sand a more clear message and more easily understandable. Maybe "Worse sleep quality was found in students who are less than 20 years of age (p = 0.007), who are sophomores (p = 0.034), who participated in part-time jobs (p = 0.022), who did not engage in leisurely sports (p = 0.008), who had relatively low perceived employment prospects (p = 0.009), who had no hobbies (p = 0.014), and who's preferred coping strategies is to ignore the problems (p < 0.001) (Table 1)." It is more direct and understandable with the object in the first part of the sentence.
We greatly appreciate your patience, and we have made changes according to your advice.
Line193 – 198, page 4: Worse sleep quality was found in students who are less than 20 years of age (p = 0.007), who are sophomores (p = 0.034), who participated in part-time jobs (p = 0.022), who did not engage in leisurely sports (p = 0.008), who had relatively low perceived employment prospects (p = 0.009), who had no hobbies (p = 0.014), and who's preferred coping strategies is to ignore the problems (p < 0.001) (Table 1).
- Line 199: there is a typo, please insert the point (.) after figure 2B.
We have revised these two errors and double-checked the rest of the article for similar errors.
Line 216 – 217, page 7: 3.3. Structural Equation Model: Direct, Indirect, and Total Effects of Sleep Quality on Self-Rated Health through Mental Health Symptom
Line 219 – 220, page 7: Standardized regression coefficients between sleep quality, depressive and anxiety symptoms, and self-rated health are displayed in Figure 2B.

Round 2
Reviewer 1 Report
The article is quite qualified to be published